# Magnitude Invariant Parametrizations Improve Hypernetwork Learning

**Jose Javier Gonzalez Ortiz**
MIT CSAIL
Cambridge, MA
josejg@mit.edu

**John Guttag**
MIT CSAIL
Cambridge, MA
guttag@mit.edu

**Adrian V. Dalca**
MIT CSAIL & MGH, HMS
Cambridge, MA
adalca@mit.edu

## Abstract

Hypernetworks, neural networks that predict the parameters of another neural network, are powerful models that have been successfully used in diverse applications from image generation to multi-task learning. Unfortunately, existing hypernetworks are often challenging to train. Training typically converges far more slowly than for non-hypernetwork models, and the rate of convergence can be very sensitive to hyperparameter choices. In this work, we identify a fundamental and previously unidentified problem that contributes to the challenge of training hypernetworks: a magnitude proportionality between the inputs and outputs of the hypernetwork. We demonstrate both analytically and empirically that this can lead to unstable optimization, thereby slowing down convergence, and sometimes even preventing any learning. We present a simple solution to this problem using a revised hypernetwork formulation that we call Magnitude Invariant Parametrizations (MIP). We demonstrate the proposed solution on several hypernetwork tasks, where it consistently stabilizes training and achieves faster convergence. Furthermore, we perform a comprehensive ablation study including choices of activation function, normalization strategies, input dimensionality, and hypernetwork architecture; and find that MIP improves training in all scenarios. We also provide easy-to-use code that can turn existing networks into MIP-based hypernetworks.

## 1 Introduction

Hypernetworks, neural networks that predict the parameters of another neural network, are increasingly important models in a wide range of applications such as Bayesian optimization (Krueger et al., 2017; Pawlowski et al., 2017), generative models (Alaluf et al., 2022; Zhang & Agrawala, 2023), amortized model learning (Bae et al., 2022; Dosovitskiy & Djolonga, 2020; Hoopes et al., 2022), continual learning (Ehret et al., 2021; von Oswald et al., 2020), multi-task learning (Mahabadi et al., 2021; Serrà et al., 2019), and meta-learning (Bensadoun et al., 2021; Zhao et al., 2020). Despite their advantages and growing use, training hypernetworks is challenging. Compared to non-hypernetwork-based models, training existing hypernetworks is often unstable. At best this increases training time, and at worst it can prevent training from converging at all. This burden limits their adoption, negatively impacting many applications. Existing hypernetwork heuristics, like gradient clipping (Ha et al., 2016; Krueger et al., 2017), are most often insufficient, while existing techniques to improve standard neural network training often fail when applied to hypernetworks.

This work addresses a cause of training instability. We identify and characterize a previously unstudied hypernetwork design problem and provide a straightforward solution to address it. We demonstrate analytically and empirically that the typical choices of architecture and parameter initialization in hypernetworks cause a proportionality relationship between the scale of hypernetwork inputs and the scale of parameter outputs (Fig. 1a). The resulting fluctuations in predicted parameter scale lead to large variability in the scale of gradients during optimization, resulting in unstable training and slow convergence. In some cases, this phenomenon prevents any meaningful learning. To overcome the identified magnitude proportionality issue, we propose a revision to hypernetwork models: Magnitude Invariant Parametrizations (MIP). MIP effectively eliminates the influence of the scale of hypernetwork inputs on the scale of the predicted parameters, while retaining the representational power of existing formulations. We demonstrate the proposed solution across several hypernetwork

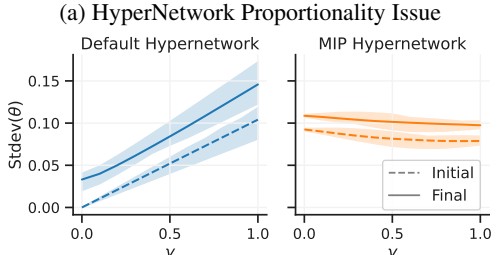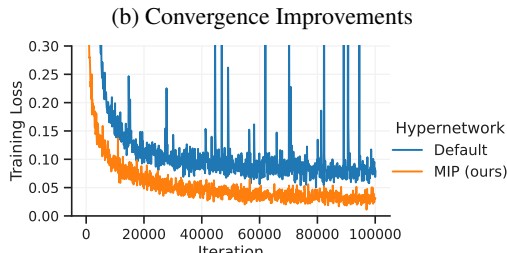

Figure 1: **(a) Proportionality Issue**. With *default* formulations, the scale of the predicted parameters $\theta$ (measured in standard deviation) is directly proportional to scale of the hypernetwork input $\gamma$ at initialization (initial), and even after training the model (final). Our proposed Magnitude Invariant Parametrizations (MIP) mitigates this proportionality issue with respect to $\gamma$. **(b) Convergence Improvements**. Using MIP leads to faster convergence and results in more stable training than the default hypernetwork formulation. The latter experiences sporadic training instabilities (spikes in the training loss).

learning tasks, providing evidence that hypernetworks using MIP achieve faster convergence and more stable training than typical hypernetwork formulation (Fig. 1b).

Our main contributions are:

- We characterize a previously unidentified optimization problem in hypernetwork training, and show that it leads to large gradient variance and unstable training dynamics.

- We propose a solution: Magnitude Invariant Parametrizations (MIP), a hypernetwork formulation that addresses the issue without introducing additional training or inference costs.

- We rigorously study the proposed parametrization. We first compare it with the standard formulation and against popular normalization strategies, showing that it consistently leads to faster convergence and more stable training. We then extensively test it using various choices of optimizer, input dimensionality, hypernetwork architecture, and activation function, finding that it improves hypernetwork training in all evaluated settings.

- We release our implementation as an open-source PyTorch library, HyperLight. Hyper-Light facilitates the development of hypernetwork models and provides principled choices for parametrizations and initializations, making hypernetwork adoption more accessible. We also provide code that enables using MIP seamlessly with existing models.

## 2 RELATED WORK

**Parameter Initialization**. Deep neural networks experience unstable training dynamics in the presence of exploding or vanishing gradients (Goodfellow et al., 2016). Weight initialization plays a critical role in the magnitude of gradients, particularly during the early stages of training. Commonly, weight initialization strategies focus on preserving the magnitude of activations during the forward pass and maintaining the magnitude of gradients during the backward pass (Glorot & Bengio, 2010; He et al., 2015). In the context of hypernetworks, early work made use of Glorot and Kaiming initialization (Balažević et al., 2019; Pawlowski et al., 2017), while more recent work proposes initializations that accounts for the architectural properties of the primary network (Beck et al., 2023; Chang et al., 2019; Knyazev et al., 2021; Zhmoginov et al., 2022). However, most of these works assume hypernetwork inputs to be categorical embeddings, which limits their applicability, and makes their formulations susceptible to the proportionality issue that we identify in this work.

**Normalization Techniques**. Normalization techniques control the distribution of weights and activations, often leading to improvements in convergence by smoothing the loss surface (Bjorck et al., 2018; Ioffe, 2017; Lubana et al., 2021; Santurkar et al., 2018). Batch normalization is widely used to normalize activations using minibatch statistics, and methods like layer or group normalization instead normalize across features (Ba et al., 2016; Ulyanov et al., 2016; Wu & He, 2018). Other methods reparametrize the weights using weight-normalization strategies to decompose direction

and magnitude (Qiao et al., 2019; Salimans & Kingma, 2016). As we show in our experiments, these strategies fail to resolve the proportionality issue we study. They either maintain the proportionality relationship, or eliminate proportionality by rendering the predicted weights independent of the hypernetwork input, eliminating the utility of the hypernetwork itself.

**Adaptive Optimization**. High gradient variance can be detrimental to model convergence in stochastic gradient methods (Johnson & Zhang, 2013; Roux et al., 2012). Solutions to mitigate this issue encompass adaptive optimization techniques, which aim to decouple the effect of gradient direction and magnitude by normalizing by a history of gradient magnitudes (Kingma & Ba, 2014; Zeiler, 2012). Similarly, applying momentum reduces the instantaneous impact of stochastic gradients by using parameter updates based on an exponentially decaying average of past gradients (Nesterov, 2013; Qian, 1999). These strategies are implemented by many widely-used optimizers, such as Adam (Balles & Hennig, 2018; Kingma & Ba, 2014). We show experimentally that although adaptive optimizers like Adam enhance hypernetwork optimization, they do not address the root cause of the identified proportionality issue.

**Fourier Features**. High-dimensional Fourier projections have been used in feature engineering (Rahimi et al., 2007) and for positional encodings in language modeling applications to account for both short and long range relationships (Su et al., 2021; Vaswani et al., 2017). Additionally, implicit neural representation models benefit from sinusoidal representations (Sitzmann et al., 2020; Tancik et al., 2020). Our work also uses low dimensional Fourier projections. We demonstrate their use as a means to project hypernetwork inputs to a vector space with constant Euclidean norm.

**Residual Forms**. Residual and skip connections are widely used in deep learning models and often improve model training, particularly with increasing network depth (He et al., 2016a;b; Li et al., 2018; Vaswani et al., 2017). Building on this intuition, instead of the hypernetworks predicting the network parameters directly, our proposed hypernetworks predict parameter *changes*, mitigating part of the proportionality problem at hand.

## 3 THE HYPERNETWORK PROPORTIONALITY PROBLEM

**Preliminaries**. Deep learning tasks generally involve a model $f(x; \theta) \rightarrow y$, with learnable parameters $\theta$. In hierarchical models using hypernetworks, the parameters $\theta$ of the *primary network* $f$ are predicted by a *hypernetwork* $h(\gamma; \omega) \rightarrow \theta$ based on a input vector $\gamma$. Instead of learning parameters $\theta$ of the primary network $f$, only the learnable parameters $\omega$ of the hypernetwork $h$ are optimized using backpropagation. The specific nature of the hypernetwork inputs $\gamma$ varies across applications, but regularly corresponds to a low dimensional quantity that models properties of the learning task, and is often a simple scalar or embedding vector (Dosovitskiy & Djolonga, 2020; Hoopes et al., 2022; Lorraine & Duvenaud, 2018; Ukai et al., 2018; Wang et al., 2021).

**Assumptions**. For analysis, we assume the following about the hypernetwork formulation: 1) The architecture is a series of fully connected layers $\phi(Wx + b)$ where $W$ are the parameters, $b$ the biases and $\phi(x)$ the non-linear activation function; 2) The nonlinear activation is a piece-wise linear function with a single switch at the origin. Namely, it satisfies $\phi(x) = \mathbb{1}_{[x \geq 0]}(\alpha x) + \mathbb{1}_{[x < 0]}(\beta x)$, for $\alpha, \beta > 0$ (e.g., LeakyReLU) 3) Bias vectors $b$ are initialized to zero. Existing hypernetworks satisfy these properties for the large majority of applications (Dosovitskiy & Djolonga, 2020; Ha et al., 2016; Lorraine & Duvenaud, 2018; MacKay et al., 2019; Ortiz et al., 2023; Ukai et al., 2018; von Oswald et al., 2020; Wang et al., 2021).

**Input-Output Proportionality**. We demonstrate that under these widely-used settings, hypernetwork inputs and outputs involve a proportionality relationship, and describe how this can impede hypernetwork training. We show that 1) at initialization, any intermediate feature vector $x^{(k)}$ at layer $k$ will be proportional to hypernetwork input $\gamma$, even under the presence of non-linear activation functions, and 2) this leads to large gradient magnitude fluctuations detrimental to optimization.

We first consider the case where $\gamma \in \mathbb{R}$ is a scalar value. Let $h(\gamma; \omega)$ use a fully connected architecture composed of a series of fully connected layers

$$h(\gamma; \omega) = W^{(n)}x^{(n)} + b^{(n)}$$
$$x^{(k+1)} = \phi(W^{(k)}x^{(k)} + b^{(k)}) \tag{1}$$
$$x^{(1)} = \gamma$$

where $x^{(k)}$ is the input vector of the $k^{\text{th}}$ fully connected layer with learnable parameters $W^{(k)}$ and biases $b^{(k)}$. To prevent gradients from exploding or vanishing when chaining several layers, it is common to initialize the parameters $W^{(i)}$ and biases $b^{(i)}$ so that either the magnitude of the activations is approximately constant across layers in the forward pass (known as *fan in*), or so that the magnitude of the gradients is constant across layers in the backward pass (known as *fan out*) (Glorot & Bengio, 2010; He et al., 2015). In both settings, the parameters $W^{(i)}$ are initialized using a zero mean Normal distribution and bias vectors $b^{(i)}$ are initialized to zero. If $\gamma > 0$, and $\phi(x)$ has the common form specified above, at initialization the $i^{\text{th}}$ entry of vector $x^{(2)}$ is

$$x_i^{(2)} = \phi(W_i^{(1)}\gamma + b^{(1)}) = \gamma\phi(W_i^{(1)}) \propto \gamma, \tag{2}$$

since $b^{(1)} = 0$ and $\phi(W_i^{(1)})$ is independent of $\gamma$. Using induction, we assume that for layer $k$, $x_j^{(k)} \propto \gamma \; \forall j$, and show this property for layer $k+1$. The value of the $i^{\text{th}}$ element of vector $x^{(k+1)}$ is

$$x_i^{(k+1)} = \phi\left(b_i^{(k)} + \sum_j W_{ij}^{(k)}x_j^{(k)}\right) = \gamma\,\phi\left(\sum_j W_{ij}^{(k)}\alpha_j^{(k)}\right) \propto \gamma, \tag{3}$$

since $b_i^{(k)} = 0$, and the term inside $\phi$ is independent of $\gamma$. If $\gamma$ is not strictly positive, we can reach the same proportionality result, but with separate constants for the positive and the negative range. This dependency holds regardless of the number of layers and the number of neurons per hidden layer, and also holds when residual connections are employed. When $\gamma$ is a vector input, we find a similar relationship with the overall magnitude of the input and the magnitude of the output. Given the absence of bias terms, and the lack of multiplicative interactions in the architecture, the fully connected network propagates magnitude changes in the input.

**Training implications**. Since $\theta = x^{(n+1)}$, this result leads to a proportionality relationship for the magnitude of the predicted parameters $||\theta||_2 \propto ||\gamma||$ and their variance $\text{Var}(\theta) \propto ||\gamma||^2$. As the scale of the primary network parameters $\theta$ will depend on $\gamma$, this will affect the scale of the layer outputs and gradients of the primary network. In turn, these large gradient magnitude fluctuations lead to unstable training dynamics for stochastic gradient descent methods (Glorot & Bengio, 2010).

**Further Considerations**. Our analysis relies on biases being at zero, which only holds at initialization, and does not include normalization layers that are sometimes used. However, in our experiments, we find that biases remain near zero during early training, and hypernetworks with alternative choices of activation function, input dimensionality, or with normalization layers, still suffer from the identified issue and consistently benefit from our proposed parametrization (see Section 6).

## 4 MAGNITUDE INVARIANT PARAMETRIZATIONS

To address the proportionality dependency, we make two straightforward changes to the typical hypernetwork formulation: 1) We introduce an encoding function that maps inputs into a constant-norm vector space, and 2) we treat hypernetwork predictions as additive *changes* to the main network parameters, rather than as the parameters themselves. These changes make the primary network weight distribution non-proportional to the hypernetwork input and stable across the range of hypernetwork inputs. Figure 2 illustrates these changes to the hypernetwork.

**Input Encoding.** To address the proportionality problem, we map the inputs $\gamma \in [0,1]$ to a space with a constant Euclidean norm $||\text{E}_{\text{L2}}(\gamma)||_2 = 1$ using the function $\text{E}_{\text{L2}}(\gamma) = [\cos(\gamma\pi/2), \sin(\gamma\pi/2)]$. With this change, the input magnitude to the hypernetwork is constant, so $||x^{(1)}|| \not\propto \gamma$. For higher-dimensional inputs, we apply this transformation to each input individually, leading to an output vector with double the number of dimensions. This transformation results in an input representation with a constant norm, thereby eliminating the proportionality effect.

For our input encoding, we first map each dimension of the input vector to the range $[0,1]$ to maximize output range of $\text{E}_{\text{L2}}$. We use min-max scaling of the input: $\gamma' = (\gamma - \gamma_{\text{min}})/(\gamma_{\text{max}} - \gamma_{\text{min}})$. For unconstrained inputs, such as Gaussian variables, we first apply the logistic function $\sigma(x) = 1/(1 + \exp(-x))$. If inputs span several orders of magnitude, we take the log before the min-max scaling as in (Bae et al., 2022; Dosovitskiy & Djolonga, 2020).

**Output Encoding.** Residual forms have become a cornerstone in contemporary deep learning architectures (He et al., 2016a; Li et al., 2018; Vaswani et al., 2017). Motivated by these methods, we

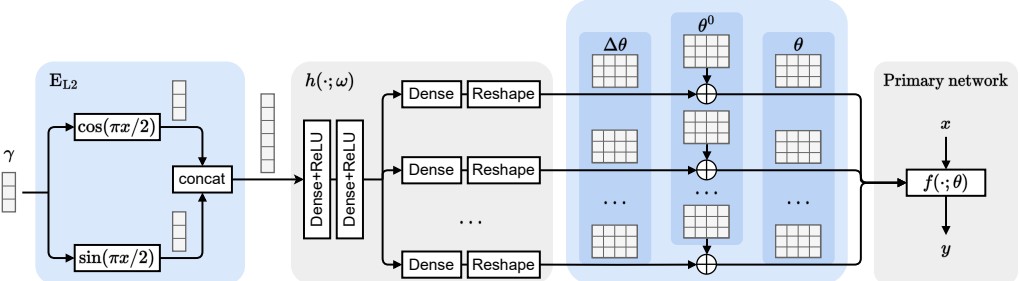

Figure 2: **Magnitude Invariant Parametrizations for Hypernetworks**. MIP first projects the hypernetwork inputs $\gamma$ to a constant norm vector space. Then the outputs of the hypernetwork $\Delta\theta$ are treated as additive changes to a set of independent learnable parameters $\theta^0$ to generate the primary network weights $\theta$. In blue we highlight the main components of MIP, the input encoding $\mathrm{E}_{\mathrm{L2}}$ and the residual formulation $\theta = \theta^0 + \Delta\theta$.

replace the standard hypernetwork framework with one that learns both primary network $f$ parameters (as is typically learned in existing formulations) *and* hypernetwork predictions, which are used as *additive* changes to these primary parameters. We introduce a set of learnable parameters $\theta^0$, and compute the primary network parameters as $\theta = \theta^0 + h(\mathrm{E}_{\mathrm{L2}}(\gamma); \omega)$.

**Parameter Initialization** We initialize the hypernetwork weights $\omega$ using common initialization methods for fully connected layers that consider the number of input and output neurons to each layer, such as Kaiming or Glorot initialization. Then, we initialize the independent parameters $\theta^0$ in the same manner that we would initialize the parameters of an equivalent regular network. We provide further details and examples in section A of the supplement.

## 5 Experimental Setup

### 5.1 Tasks

We evaluate our proposed parametrization on several tasks involving hypernetwork-based models.

**Bayesian Neural Networks**. Hypernetwork models have been used to learn families of functions conditioned on a prior distribution (Ukai et al., 2018). During training, the prior representation $\gamma \in \mathbb{R}^d$ is sampled from the prior distribution $\gamma \sim p(\gamma)$ and used to condition the hypernetwork $h(\gamma; \omega) \rightarrow \theta$ to predict the parameters of the primary network model $f(x; \theta)$. Once trained, the family of posterior networks is then used to estimate parameter uncertainty or to improve model calibration. For illustrative purposes we first evaluate a setting where $f(x; \theta)$ is a feed-forward neural network used to classify the MNIST dataset. Then, we tackle a more complex setting where $f(x; \theta)$ is a ResNet-like model trained the OxfordFlowers-102 dataset (Nilsback & Zisserman, 2006). In both settings, we use the prior $\mathcal{N}(0, 1)$ for each input.

**Hypermorph**. Learning-based medical image registration networks $f(x_m, x_f; \theta) \rightarrow \phi$ register a moving image $x_m$ to a fixed image $x_f$ by predicting a flow or deformation field $\phi$ between them. The common (unsupervised) loss balances an image alignment term $\mathcal{L}_{\mathrm{sim}}$ and a spatial regularization (smoothness) term $\mathcal{L}_{\mathrm{reg}}$. The learning objective is then $\mathcal{L} = (1-\gamma)\mathcal{L}_{\mathrm{sim}}(x_m \circ \phi, x_f) + \gamma\mathcal{L}_{\mathrm{reg}}(\phi)$, where $\gamma$ controls the trade-off. In Hypermorph (Hoopes et al., 2022), multiple regularization settings for medical image registration are learned jointly using hypernetworks. The hypernetwork is given the trade-off parameter $\gamma$ as input, sampled stochastically from $\mathcal{U}(0, 1)$ during training. We follow the same experimental setup, using a U-Net architecture for the primary (registration) network and training with MSE for $\mathcal{L}_{\mathrm{sim}}$ and total variation for $\mathcal{L}_{\mathrm{reg}}$. We train models on the OASIS dataset. For evaluation, we use the predicted flow field to warp anatomical segmentation label maps of the moving image, and measure the volume overlap to the fixed label maps (Balakrishnan et al., 2019).

**Scale-Space Hypernetworks**. We also use a hypernetwork to efficiently learn a family of models with varying internal rescaling factors in the downsampling and upsampling layers, as done in Ortiz et al. (2023). In this setting, $\gamma$ corresponds to the *scale factor*. Given hypernetwork input $\gamma$, the

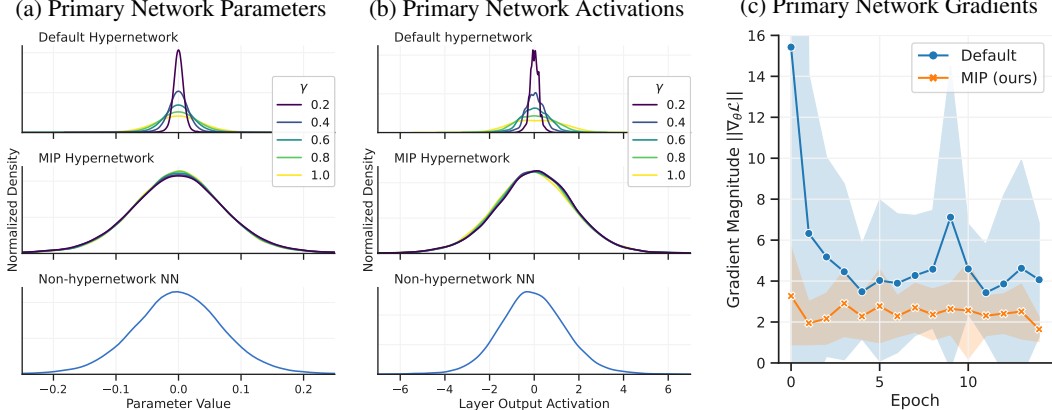

Figure 3: **Distributions of primary network parameters (a) and layer activations (b)**. Measurements are taken at initialization for a default hypernetwork, our proposed MIP hypernetwork, and a conventional neural network with the same primary architecture. Distributions are shown as kernel density estimates (KDE) of the values because of the high degree of overlap between the distributions. The MIP strategy leads to little change across input values and its distribution closely matches that of the non-hypernetwork model. **Evolution of Gradients (c)** Gradient magnitude with respect to hypernetwork outputs $||\nabla_\theta \mathcal{L}||$ during early training. Standard deviation is computed across mini-batches in the same epoch. MIP leads to substantially smaller magnitude and standard deviation compared to the default parametrization.

hypernetwork $h(\gamma; \omega) \rightarrow \theta$ predicts the parameters of the primary network, which performs the spatial rescaling operations according to the value of $\gamma$. We study a setting where $f(x; \theta)$ is a convolutional network with variable resizing layers, the rescaling factor is sampled from $\mathcal{U}(0, 0.5)$, and evaluate using the OxfordFlowers-102 classification problem and the OASIS segmentation task.

## 5.2 EXPERIMENT DETAILS

**Model**. We implement the hypernetwork as a neural network with fully connected layers and LeakyReLU activations for all but the last layer, which has linear output. Hypernetwork weights are initialized using Kaiming initialization on *fan out* mode and biases are initialized to zero. Unless specified otherwise, the hypernetwork architecture has two hidden layers with 16 and 128 neurons respectively. We use this implementation for both the default (existing) hypernetworks, and our proposed (MIP) hypernetworks.

**Training**. We use two popular choices of optimizer: SGD with Nesterov momentum, and Adam. We search over a range of initial learning rates and report the best performing models; further details are included in section B of the supplement.

**Implementation.** An important contribution of our work is HyperLight, our PyTorch hypernetwork framework. HyperLight implements the proposed hypernetwork parametrization, but also provides a modular and composable API that facilitates the development of hypernetwork models. Using HyperLight, practitioners can employ existing non-hypernetwork model definitions and pretrained model weights, and can easily build models using hierarchical hypernetworks. Anonymized source code is available at `https://github.com/anonresearcher8/hyperlight`.

## 6 EXPERIMENTAL RESULTS

### 6.1 EFFECT OF PROPORTIONALITY ON PARAMETER AND GRADIENT DISTRIBUTIONS

First, we empirically show how the proportionality phenomenon affects the distribution of predicted weights $\theta$ and their corresponding gradients for the Bayesian neural networks on MNIST. Figures 3a

and 3b compare the distributions of the primary network weights and layer outputs for a range of values of hypernetwork input $\gamma$. The default hypernetwork parametrization is highly sensitive to changes in the input, in contrast, MIP eliminates this dependency, with the resulting distribution closely matching that of the non-hypernetwork models. Figure 1a (in the introduction), shows that using the default formulation, the scale of the weights correlates linearly with the value of the hypernetwork input, and that, crucially, this correlation is still present after the training process ends. In contrast, MIP parametrizations lead to a weight distribution that is robust to the input $\gamma$, both at the start and end of training.

We also analyze how the proportionality affects the early phase of hypernetwork optimization by studying the distribution of gradient norms during training. Figure 3c shows the norm of the predicted parameter gradients $||\nabla_\theta \mathcal{L}||$ as training progresses. Consistent with our analysis, hypernetworks with default parametrization experience large swings in gradient magnitude because of the proportionality relationship between inputs and predicted parameters. In contrast, the MIP strategy leads to a substantially smaller variance and more stable gradient magnitude.

## 6.2 MODEL TRAINING IMPROVEMENTS

In this experiment, we analyze how MIP affects model convergence for the considered tasks. For all experiments, we found that MIP hypernetworks did not introduce a measurable impact in training runtime, so we report per-epoch steps.

Figure 1b (in the introduction) shows the training loss for Bayesian networks trained on MNIST. We find that MIP parametrizations result in smaller loss sooner during training, and the default parametrization suffers from sporadic training instabilities (spikes in the training loss), while MIP leads to stable training. Similarly, Figure 4a shows the test accuracy for Bayesian networks trained on OxfordFlowers. In this task, MIP also achieves faster convergence and better final model accuracy for both choices of optimizer.

Figures 4b and 4c present convergence curves for the other two tasks. For Hypermorph, MIP parametrizations are crucial when using SGD with momentum since otherwise the model fails to meaningfully train. For all choices of learning rate the default hypernetwork failed to converge, whereas with MIP parametrization it converged for a large range of values. With Adam, networks train meaningfully, and MIP models consistently achieve similar Dice scores substantially faster. They are less sensitive to weight initializations. Though the Adam optimizer partially mitigates the gradient variance issue by normalizing by a history of previous gradients, the MIP parametrization leads to substantially faster convergence. Furthermore, for the Scale-Space segmentation, we find that for both optimizers MIP models achieve substantially faster convergence and better final accuracy compared to those with the default parametrization.

**Comparison to normalization strategies.** We compare the proposed parametrization to popular choices of normalization layers found in the deep learning literature. Using the default formulation, where the predicted weights start proportional to the hypernetwork input, we found that existing normalization strategies fall into two categories: they either keep the proportionality relationship present (such as batch normalization), or remove the proportionality by making the predicted weights independent of the hypernetwork input (such as layer or weight normalization). We provide further details in Section C of the supplemental material.

We test several of these normalization strategies. **BatchNorm-P**, adds batch normalization layers to the primary network. **LayerNorm-P**, adds feature normalization layers to the primary network. **LayerNorm-H**, adds feature normalization layers to the hypernetwork layers. **WeightNorm**, performs weight normalization, which decouples the gradient magnitude and direction, to weights predicted by the hypernetwork (Ba et al., 2016; Ioffe, 2017; Salimans & Kingma, 2016). Figure 5a shows the evolution of the test accuracy for the Scale-Space hypernetworks trained on OxfordFlowers. We report wall clock time, since some normalization strategies, such as BatchNorm, substantially increase the computation time required per iteration. For networks trained with SGD, these normalization strategies enable training, but do not significantly improve on default hypernetworks when trained with Adam. Models trained with SGD momentum and hypernetwork feature normalization (LayerNorm-H) diverged early into training for all considered hyperparameter settings. Models trained with the proposed MIP parametrization lead to substantially faster convergence and better final model accuracy.

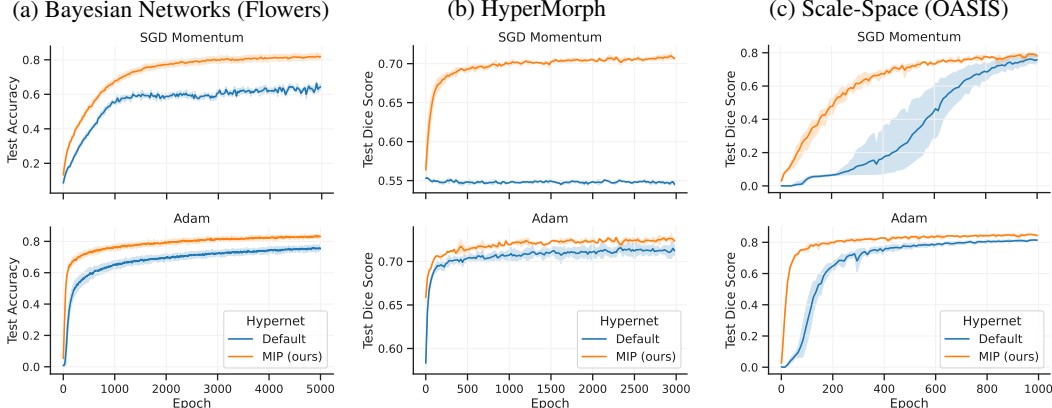

Figure 4: **Models Convergence Improvements**. Comparison between default hypernetworks and hypernetworks with MIP for the Bayesian networks on OxfordFlowers-102 (a), HyperMorph (b) and Scale-Space hypernetworks trained on OASIS (c). In all cases, we find that the MIP parametrization leads to faster model convergence without any sacrifice in final model accuracy compared to the default parametrization. In all cases we observe that MIP improves convergence and final model accuracy. We also find that for default hypernetworks using the Adam optimizer substantially helps the training process, however, incorporating MIP leads to even better training dynamics.

**Initialization Schemes.** We compare MIP and the default hypernetworks to hypernetworks that use the Hyperfan-in and Hyperfan-out initialization strategies from Chang et al. (2019). The Hyperfan initialization takes into account the hypernetwork and primary network architectures when initializing model weights, improving model convergence and training stability. However, Hyperfan initializations are not designed for magnitude-encoded inputs, so they are susceptible to the proportionality issue we identify.

Figure 5b presents convergence results for the Hypermorph task with SGD and Adam. We find that Hyperfan initializations do not resolve the training challenges when using SGD. For hypernetworks trained with Adam, MIP outperforms both Hyperfan variants.

**Ablation Analysis.** We study the contribution of each of the two main components of the MIP parametrizations: input encoding and additive output formulation. Figure 5c shows the effect on convergence for two tasks. We found that both components reduce the proportionality dependency between the hypernetwork inputs and outputs, and that each component independently achieves substantial improvements in model convergence. However, we find that best results (fastest convergence) are consistently achieved when both components are used jointly during training.

### 6.3 ROBUSTNESS ANALYSIS

**Hypernetwork Input Dimensionality**. We study the effect of the number of dimensions of the input to the hypernetwork model. We evaluate on the Bayesian neural network task, and we vary the number of dimensions of the input prior. We train models with geometrically increasing number of input dimensions, $\dim(\gamma) = 1, 2, \ldots, 32$. Figure 6 (in section C.1 of the supplement) shows that the proposed MIP strategy leads to improvements in model convergence and final model accuracy as we increase the dimension of the hypernetwork input $\gamma$.

**Choice of Hypernetwork Architecture.** We assess model performance when varying the properties of the hypernetwork architecture. We vary the width (number of hidden neurons per layer) and depth (number of layers)– fully connected networks with 3, 4 and 5 layers and with 16 and 128 neurons per layer, as well as an exponentially growing number of neurons per layer $\dim(x^n) = 16 \cdot 2^n$. Figures 7 and 8 (in section C.2 of the supplement) show that the MIP improvements generalize to the all tested hypernetwork architectures with analogous improvements in model training.

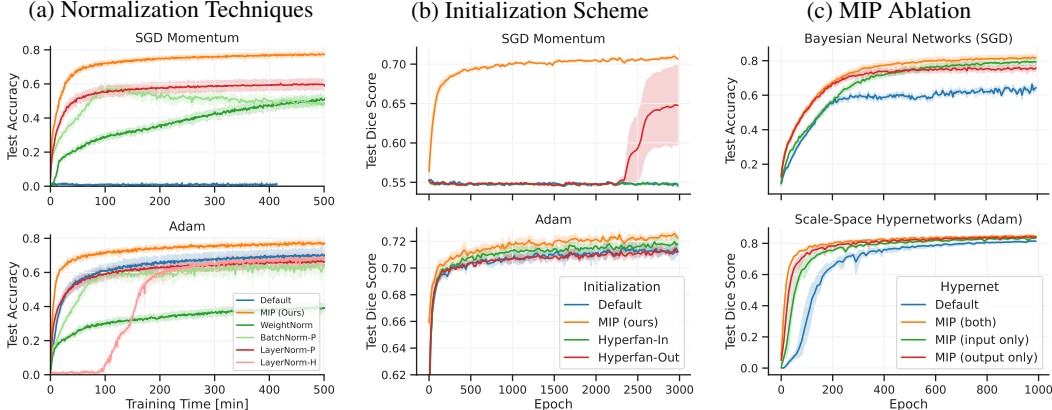

Figure 5: **(a) Normalization Strategies**. Comparison of hypernetworks trained with various normalization strategies for Scale-Space hypernetworks trained of OxfordFlowers-102. MIP provides substantially better results than the considered normalization strategies, achieving faster model convergence and better final test accuracy. **(b) Initialization Scheme.** Comparison of hypernetworks with the default, MIP, Hyperfan-In and Hyperfan-Out initialization schemes on the Hypermorph task. Models with MIP train substantially better than models with Hyperfan initializations, especially when using the SGD optimizer. **(c) Ablation Analysis.** Convergence results for separate components of MIP on the Scale-Space Hypernetworks on OASIS using the Adam optimizer, and on the Bayesian networks trained with SGD on the OxfordFlowers-102 classification problem. Each component of the parametrization leads to improvements in final model accuracy as well as training convergence, and best results are achieved when using both components simultaneously.

**Choice of Nonlinear Function Activation**. While our method is motivated by the training instability present in hypernetworks with (Leaky)-ReLU nonlinear activation functions, we explored applying it to other common choices of activation functions found in the literature: Tanh, GELU and SiLU (Hendrycks & Gimpel, 2016; Ramachandran et al., 2017). Figure 9 (in section C.3 of the supplement) shows that MIP consistently helps for all choices of nonlinear activation function, and the improvements are similar to those of the LeakyReLU models.

## 7   LIMITATIONS

All hypernetwork models used in our experiments are composed of fully connected layers and use activation and initialization choices commonly recommended in the literature. Similarly, we focused on two optimizers in our experiments, SGD with momentum and Adam. We believe that we would see similar results for other less common architectures and optimizers, but this remains to be investigated. Furthermore, we focus on training models from scratch. As hypernetworks become popular in transfer learning, we believe this will be an interesting avenue for future analysis of MIP.

## 8   CONCLUSION

We showed through analysis and experimentation that traditional hypernetwork formulations are susceptible to training instability, caused by the effect of the magnitude of hypernetwork input values on primary network weights and gradients, and that standard methods such as batch and layer normalization do not solve the problem. We then proposed the use of a new method, Magnitude Invariant Parametrizations (MIP), for addressing this problem. Through extensive experiments, we demonstrated that MIP leads to substantial improvements in convergence times and model accuracy across multiple hypernetwork architectures, training scenarios, and tasks. Given that using MIP never reduces model performance and can dramatically improve training, we expect the method to be widely useful for training hypernetworks.

## ACKNOWLEDGEMENTS

This research is supported by Quanta Computer, Inc, and Wistron Coporation. Additionally, the project was supported by NIH R01AG064027 and R01AG070988.

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

APPENDIX

## A  MIP PARAMETER INITIALIZATION

Parameter initialization strategies for regular neural networks have remained fairly stable over the past half decade. The prevalent method consists of using Glorot or He initializations that are designed to preserve the magnitude of activations during the forward pass and maintain the magnitude of gradients during the backward pass (Glorot & Bengio, 2010; He et al., 2015).

Typically, the assumptions of these initialization schemes do not hold when applied to hypernetwork settings. This has prompted the development of several specialized initialization schemes tailored for hypernetwork formulations (Beck et al., 2023; Chang et al., 2019; Knyazev et al., 2021). Crucially, these techniques require incorporating the knowledge of the primary network when performing the initialization, and are designed for categorical inputs represented as embedding vectors. The guarantees these schemes provide do not hold when using magnitude-encoded inputs such as scalars.

We propose a simple yet effective initialization scheme based on the recommendations from the neural network literature. First, the hypernetwork weights $\omega$ are initialized using common initialization methods for fully connected layers. Then, the independent parameters $\theta^0$ are initialized taking into consideration their role in the primary network.

We illustrate our initialization rules using examples with the Kaiming He *fan-out* scheme. Using He fan-out init, the weights and biases of a neural network layer with $n_{\text{in}}$ input neurons and $n_{\text{out}}$ output neurons are sampled

$$W \sim \mathcal{N}\left(0, \frac{G}{\sqrt{n_{\text{out}}}}\right) \qquad b = 0, \tag{4}$$

where $G$ is the relative *gain* of the non-linearity function $\phi(x)$. For ReLU, we have $G = 2$ whereas linear or sigmoid, we have $G = 1$.

We differentiate the following cases:

- **Intermediate Hypernetwork Layer** - We initialize a $(n_{\text{in}}, n_{\text{out}})$ layer in the hypernetwork following the scheme we just outlined, i.e., $W \sim \mathcal{N}(0, G/(\sqrt{n_{\text{out}}})$ and $b = 0$.

- **Final Hypernetwork Layer** – We consider two cases, but we do not consider biases because they are redundant with $\theta_0$.

  1. Layer predicting a primary network weight of shape $(n_{\text{in}}, n_{\text{out}})$:

  $$W \sim \mathcal{N}\left(0, \frac{1}{\sqrt{n_{\text{in}} n_{\text{out}}}}\right) \tag{5}$$

  2. Layer predicting a primary network bias: $W = 0$

- **Independent Weights** $\theta_0$ – For fully connected layer with $(n_{\text{in}}, n_{\text{out}})$ neurons, we initialize $W \sim \mathcal{N}(0, G/(\sqrt{n_{\text{out}}})$, and $b = 0$.

From this initialization, we can observe that the set of independent weights $\theta_0$ is initialized as if they were the weights of a regular neural network. Alternatively, if the primary network corresponds to a pretrained model, the independent weights $\theta_0$ are initialized using the pretrained values, and can be optionally frozen during training.

### A.1  IMPLEMENTATION CONSIDERATIONS

Since the $\theta_0$ weights are redundant with the bias parameters of the final hypernetwork layer, we remove bias parameters from the final hypernetwork layer. In our implementation, we use a single final layer, but we initialize its weights as if it were the multiple smaller layers, since otherwise the initialization would not follow the recommendations outlined in the previous section.

Under some hypernetwork configurations, all the primary network parameters are predicted in a single forward pass of the hypernetwork. In this scenario, we implement the parameters $\theta^0$ as the

bias vector terms of the last layer $b^{(n)}$, which proves to be as efficient as the default formulation. This is correct because we do not have $b^{(n)}$ in our formulation, and $b^{(n)}$ is equivalent to $\theta^0$ in the computational graph, receiving the same gradients. Hence, we initialize $b^{(n)}$ as a one-dimensional representation of the primary network parameters, subsequently reshaping it to construct $\theta$.

# B  ADDITIONAL EXPERIMENTAL DETAILS

## B.1  DATASETS

**MNIST**. We train models on the MNIST digit classification task. We use the official MNIST database of handwritten digits. The MNIST database of handwritten digits comprises a training set of 60,000 examples, and a test set of 10,000 examples. We use the official train-test split for training data, and further divide the training split into training and validation using a stratified 80%-20% split. We use the digit labels and consider the 10-way classification problem.

**OxfordFlowers-102**. We use the OxfordFlowers-102 dataset, a fine-grained vision classification dataset with 8,189 examples from 102 flower categories (Nilsback & Zisserman, 2006). We utilize this dataset as it poses a non-trivial learning task that does not quickly converge, and allows us to better study learning dynamics. We use the official train-test split for training data, and further divide the training split into training and validation using a stratified 80%-20% split. We perform data augmentation by considering random square crops of between 25% and 100% of the original image area and resizing images to 256 by 256 pixels. Additionally, we perform random horizontal flips and color jitter (brightness 25%, contrast 50%, saturation 50%). For evaluation we take the central square crop of each image and resize to 256 by 256 pixels.

**OASIS** We use a version of the open-access OASIS Brains dataset (Hoopes et al., 2022; Marcus et al., 2007), a medical imaging dataset containing 414 MRI scans from separate individuals, comprised of skull-stripped and bias-corrected images that are resampled into an affinely-aligned, common template space. For each scan, segmentation labels for 24 brain substructures in a 2D coronal slice are available. We use 64%, 16% and 20% splits for training, validation and test.

## B.2  BAYESIAN NEURAL NETWORKS

**Primary Network**. For the MNIST task, we use a LeNet architecture variant that uses ReLU activations as they have become more prevalent in modern deep learning models. Moreover, we replace the first fully-connected layer with two convolutional layers of 32 and 64 features. We found this change did not impact test accuracy in non-hypernetwork models, but it lead to more stable initializations for the default hypernetworks.

For the OxfordFlowers-102 task, the primary network $f$ features a ResNet-like architecture with five downsampling stages with (16, 32, 64, 128, 128) feature channels respectively. For experiments including normalization layers, such as BatchNorm and LayerNorm, the learnable affine parameters of the normalization layers are not predicted by the hypernetworks and are optimized like in regular neural networks via backpropagation.

**Training**. We train using a categorical cross entropy loss. For both optimizers we use learning rate $\eta = 3 \times 10^{-4}$. Nevertheless, we found consistent results with the ones we report using learning rates in the range $\eta = [10^{-4}, 3 \times 10^{-3}]$. We sample $\gamma$ from the uniform distribution $\mathcal{U}[0, 1]$.

**Evaluation**. For evaluation we use top-1 accuracy on the classification labels. In order to get a more fine-grained evolution of the test accuracy, we evaluate on test set at 0.25 epoch increments during training. We report results with five model replicas with different random seeds.

## B.3  HYPERMORPH

HyperMorph, a learning based strategy for deformable image registration learns models with different loss functions in an amortized manner. In image registration, the $\gamma$ hypernetwork input controls the trade-off between the reconstruction and regularization terms of the loss.

**Primary Network**. For our primary network $f$ we use a U-Net architecture (Ronneberger et al., 2015) with a convolutional encoder with five downsampling stages with two convolutional layers

per stage of 32 channels each. Similarly, the convolutional decoder is composed of four stages with two convolutional layers per stage of 32 channels each. We found that models with more convolutional filters performed no better than the described architecture.

**Training**. We train using the setup described in HyperMorph (Hoopes et al., 2022) using mean squared error for the reconstruction loss and total variation for the regularization of the predicted flow field. For the Adam optimizer we use $\beta_1 = 0.9$ and $\beta_2 = 0.999$ with decoupled decay Loshchilov & Hutter (2017) and $\eta = 10^{-4}$, but we found that learning rates $[10^{-4}, 3 \times 10^{-3}]$ lead to similar convergence results. For SGD with momentum, we tested learning rates $\eta = \{3 \times 10^{-2}, 10^{-2}, 3 \times 10^{-3}, 10^{-3}, 3 \times 10^{-4}, 10^{-4}, 3 \times 10^{-5}, 10^{-5}, \}$. In all cases the default hypernetwork formulation failed to meaningfully train. We train for 3000 epochs, and sample $\gamma$ uniformly in the range $[0, 1]$ like in the original work.

**Evaluation**. Like Hoopes et al. (2022), we use segmentation labels as the main means of evaluation and use the predicted flow field to warp the segmentation label maps and measure the overlap to the ground truth using the Dice score (Dice, 1945), a popular metric for measuring segmentation quality. Dice score quantifies the overlap between two regions, with a score of 1 indicating perfect overlap and 0 indicating no overlap. For multiple segmentation labels, we compute the overall Dice coefficient as the average of Dice coefficients for each label. We report results with five model replicas with different random seeds.

### B.4 SCALE-SPACE HYPERNETWORKS

We evaluate on a task where the hypernetwork input $\gamma$ controls architectural properties of the primary network. We use $\gamma$ to determine the amount of downsampling in the pooling layers. Instead of using pooling layers that rescale by a fixed factor of two, we replace these operations by a fractional bilinear sampling operation that rescales the input by a factor of $\gamma$.

**Primary Network**. For classification tasks, our primary network $f$ features a ResNet-like architecture with five downsampling stages with $(16, 32, 64, 128, 128)$ feature channels respectively. For experiments including normalization layers, such as BatchNorm and LayerNorm, the learnable affine parameters of the normalization layers are not predicted by the hypernetworks and are optimized like in regular neural networks via backpropagation.

For segmentation tasks, we model the primary network $f$ using a U-Net architecture (Ronneberger et al., 2015) with a convolutional encoder with five downsampling stages with two convolutional layers per stage of 32 channels each. Similarly, the convolutional decoder is composed of four stages with two convolutional layers per stage of 32 channels each.

**Training.** We sample the hypernetwork input $\gamma$ uniformly in the range $[0, 0.5]$ where $\gamma = 0.5$ corresponds to downsampling by 2. We train the multi-class classification task using a categorical cross-entropy loss, and train with a weight decay factor of $10^{-3}$, and with label smoothing Goodfellow et al. (2016); Szegedy et al. (2016) the ground truth labels with a uniform distribution of amplitude $\epsilon = 0.1$. For the segmentation tasks we train using a cross-entropy loss and then finetune using a soft-Dice loss term, as in Ortiz et al. (2023). For both optimizers we use learning rate $\eta = 1 \times 10^{-4}$. Nevertheless, we found consistent results with the ones we report using learning rates in the range $\eta = [1 \times 10^{-4}, 3 \times 10^{-3}]$.

## C   ADDITIONAL EXPERIMENTAL RESULTS

### C.1   NUMBER OF INPUT DIMENSIONS

In this experiment, we study the effect of the number of dimensions of the input to the hypernetwork model on the hypernetwork training process, both for the default parametrization and for our MIP parametrization. We evaluate using the Bayesian Hypernetworks, since we can vary the number of dimensions of the input prior without having to define new tasks. We train models with geometrically increasing number of input dimensions, $\dim(\gamma) = 1, 2, \ldots, 32$. We apply the input encoding to each dimension independently. We study two types of input distribution: uniform $\mathcal{U}(0, 1)$ and Gaussian $\mathcal{N}(0, 1)$. For MIP, we apply a sigmoid to the Gaussian inputs to constrain them to the [0,1] range as specified by our method. We evaluate on the Bayesian hypernetworks task on the OxfordFlowers-102 dataset with a primary convolutional network optimized with Adam.

Figure 6 shows the convergence curves during training. Results indicate that the proposed MIP parametrization leads to improvements in model convergence and final model accuracy for all number of input dimensions to the hypernetwork and for both choices of input distribution. Moreover, we observe that the gap between MIP and the default parametrization does not diminish as the number of input dimensions grows.

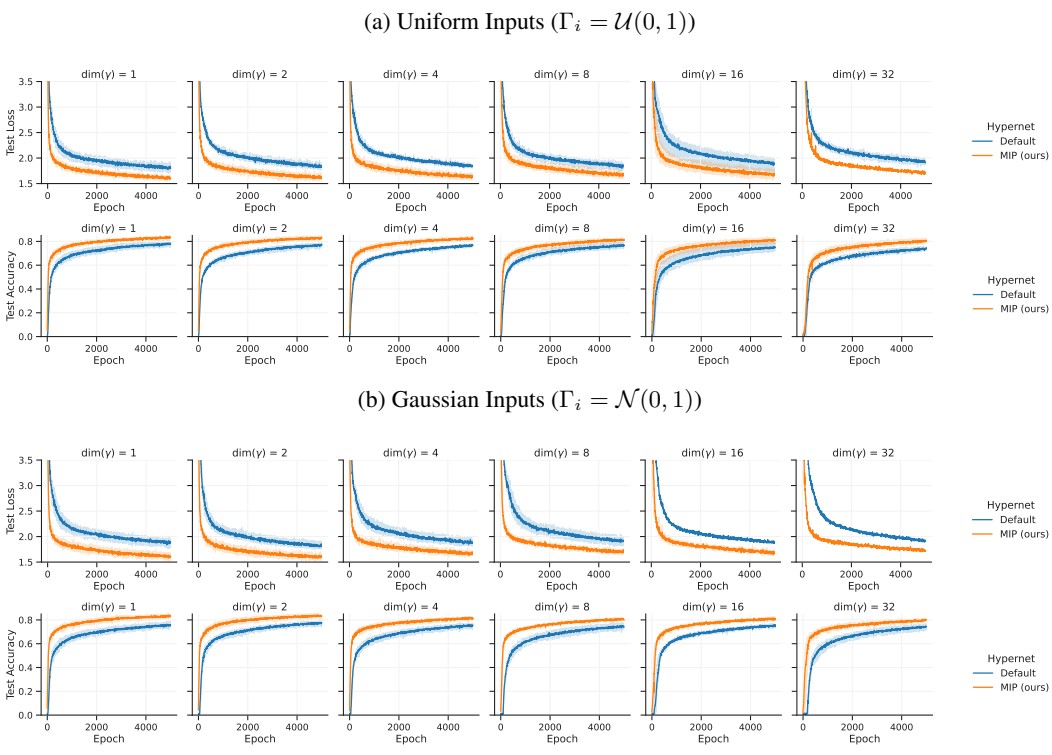

Figure 6: **Number of dimensions of hypernetwork input**. Test loss (top row) and test accuracy (bottom row) for Bayesian hypernetworks trained on the OxfordFlowers classification task for increasing number of dimensions of the hypernetwork input $\gamma$. We report results for different prior input distributions: Uniform (a) and Gaussian (b). For each setting, we train 3 independent replicas with different random initialization and report the mean (solid line) and the standard deviation (shaded region). We see significant improvements in model training convergence when the hypernetwork uses the proposed MIP parametrization.

### C.2 CHOICE OF HYPERNETWORK ARCHITECTURE

In this experiment we test whether increasing the choice of hypernetwork architecture size has an effect on the improvements achieved by incorporating Magnitude Invariant Parametrizations (MIP). We study varying the width (the number of neurons per hidden layer) and the depth (the number of hidden layers) independently as well as jointly. For the depth, we consider networks with 3, 4 and 5 layers. For width, we consider having 16 neurons per layer, 128 neurons per layer, or having an exponentially growing number of neurons per layer (exp), following the expression $\text{Dim}(x^n) = 16 \cdot 2^n$.

We compare training networks using the default hypernetwork parametrization and MIP for the HyperMorph task. Figure 7 shows convergence curves for the evaluated settings, for several random initializations. Additionally, Figure 8 shows the distribution of final model performances for the range of inputs $\gamma \in [0, 1]$. We find that MIP models converge faster without sacrificing final model accuracy.

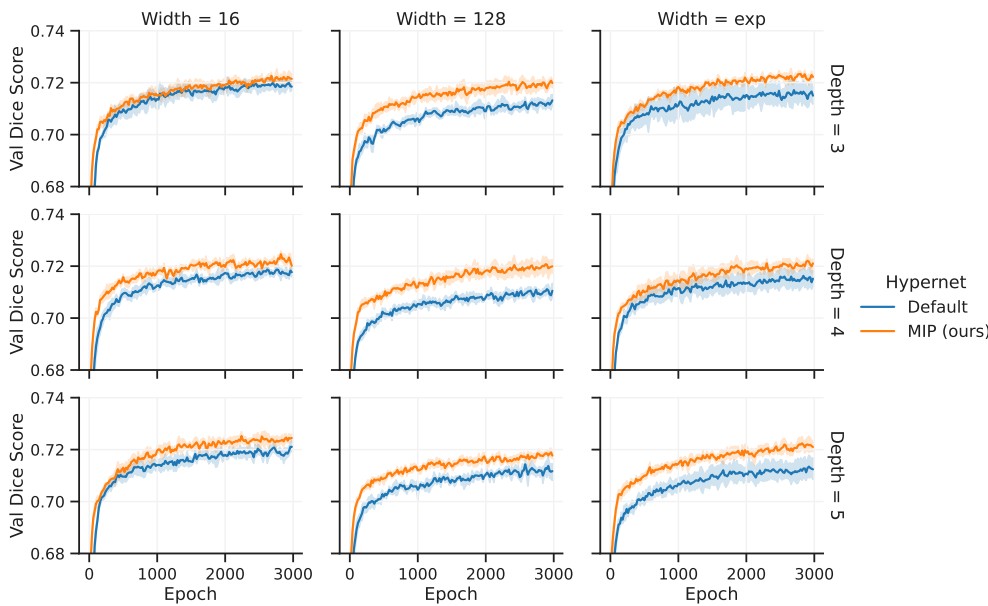

Figure 7: Model convergence for several configurations of depth and width of the hypernetwork architecture for default and MIP hypernetworks. Results are for HyperMorph on OASIS. Shaded regions measure standard deviation across hypernetwork initializations.

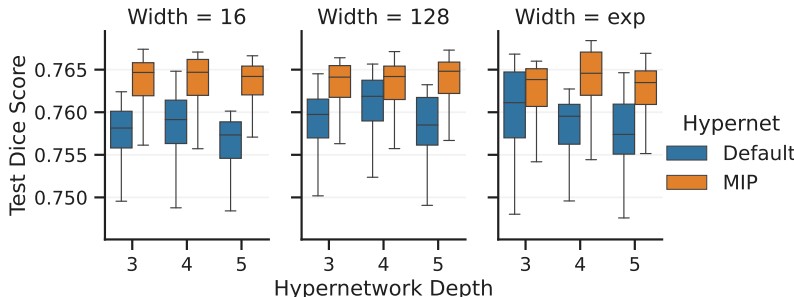

Figure 8: Test dice score for several configurations of depth and width of the hypernetwork architecture for default and MIP hypernetworks. Results are for HyperMorph on OASIS. Box-plots are reported over the range of hypernetwork inputs $\gamma$. For all hypernetwork architectures, MIP parametrizations consistently lead to more accurate models.

## C.3 CHOICE OF NONLINEAR ACTIVATION FUNCTION

While our method is motivated by the training instability present in hypernetworks with (Leaky)-ReLU nonlinear activation functions, we explored applying it to other popular choices of activation functions. We consider popular activation functions GELU and SiLU (also known as Swish) that are close to the ReLU formulation, as well as the Tanh nonlinear function Hendrycks & Gimpel (2016); Ramachandran et al. (2017).

We evaluate on the Bayesian hypernetworks task on the OxfordFlowers-102 dataset with a primary convolutional network trained optimized with Adam. Figure 9 shows the convergence curves for Bayesian hypernetworks with a primary convolutional network trained on the OxfordFlowers classification task optimized with Adam. We see that MIP consistently helps for all choices of nonlinear activation function, and the improvements are similar to those of the LeakyReLU models.

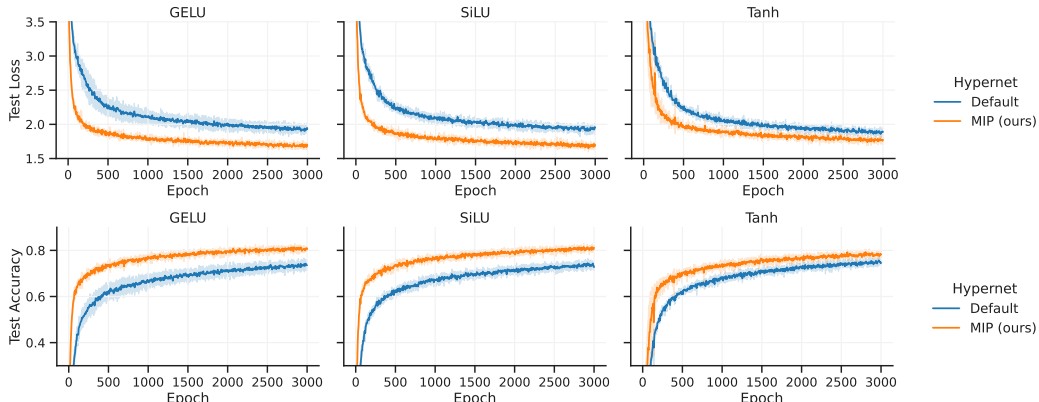

Figure 9: **MIP on alternative nonlinear activation functions** Test loss (top row) and test accuracy (bottom row) for Bayesian hypernetworks trained on the OxfordFlowers classification task for various choices of nonlinear activation function in the hypernetwork architecture: GELU, SiLU and Tanh. For each setting, we train 3 independent replicas with different random initialization and report the mean (solid line) and the standard deviation (shaded region). We see significant improvements in model training convergence when the hypernetwork uses the proposed MIP parametrization.

## C.4 FINAL MODEL PERFORMANCE

Table 1: **Final model results on the test set for the considered tasks and models.** We report the average performance averaged across the range of $\gamma$ inputs. We find MIP does not decrease model performance in any setting, while providing substantial improvements in several of them, especially when using the SGD optimizer. Standard deviation across random initializations is included in parentheses.

| | | Adam | | SGD | |
|---|---|---|---|---|---|
| **Task** | **Data** | Default | MIP | Default | MIP |
| Bayesian NN | MNIST | 98.1 (1.1) | 99.1 (0.3) | 99.2 (0.2) | 99.0 (0.2) |
| | OxfordFlowers-102 | 78.1 (1.9) | 83.2 (0.3) | 1.4 (0.1) | 75.4 (0.5) |
| HyperMorph | OASIS | 71.0 (0.3) | 72.1 (0.3) | 54.3 (0.4) | 70.5 (0.2) |
| Scale-Space HN | OASIS | 81.4 (0.3) | 84.4 (0.6) | 75.3 (2.7) | 78.8 (1.4) |

## C.5 NORMALIZATION STRATEGIES

Before developing MIP parametrizations we tested the viability of existing normalization strategies (such as Layer or Weight normalization) to deal with the identified proportionality phenomenon. While normalizing inputs and activations is a common practice in neural network training, hypernetworks present different challenges, and applying these techniques can actually be detrimental to the training process. Hypernetworks predict network parameters, and many of the assumptions behind parameter initialization and activation distribution do not easily translate between classical networks and hypernetworks.

An important distinction is that the main goal of our formulation is to ensure that the hypernetwork input has constant magnitude, not that is normalized (i.e., zero mean, unit variance). A normalized variable $z \sim \mathcal{N}(0,1)$ does not have constant magnitude (i.e., L2 norm), over its support, so normalization techniques do not solve the identified magnitude dependency and can actually lead to undesirable formulations. To show this, let $x \in \mathbb{R}^k$ be a hypernetwork activation vector, and $\gamma \in [0,1]$ the hypernetwork input. Then, according to the identified proportionality in Section 3.2, we know that $x = \gamma z$. Here $x$ is the activation when the input is $\gamma$ and $z$ is a vector independent of $\gamma$. The normalization output will be

$$\text{Norm}(x) = \frac{x - \mathbb{E}[x]}{\text{Stdev}[x]} = \frac{\gamma z - \mathbb{E}[\gamma z]}{\text{Stdev}[\gamma z]} = \frac{\gamma z - \gamma \mathbb{E}[z]}{|\gamma| \text{Stdev}[z]} = \frac{z - \mathbb{E}[z]}{\text{Stdev}[z]},$$

making the output independent of the hypernetwork input $\gamma$. Following this reasoning, strategies like layer norm, instance norm or group norm in the hypernetwork will make the output of the model independent of the hypernetwork input, rendering the hypernetwork unusable for scalar inputs. For batch normalization cases it depends upon whether different hypernetwork inputs are used for each element in the minibatch. If not, the same logic applies as in the feature normalization strategies. Otherwise, the proportionality will still hold as the batch mean and standard deviation will be the same for all entries in the minibatch. Our experimental results confirm this. Hypernetworks with layer normalization fail to train in most settings. In contrast, we found consistently that training substantially improves when using our MIP formulation. See Figure 5a in the main body which shows that none of the tested normalization strategies is competitive with MIP in terms of model convergence or final model accuracy.

**Batch Normalization** - Applying batch normalization fails to deal with the proportionality phenomenon because it normalizes statistics that are independent of the magnitude of $\gamma$ keeping the proportionality (Ioffe, 2017). In our experiments, batch normalization performed similar to the default formulation when included in either the hypernetwork or the primary network, failing to address the proportionality relationship. For instance, all of the results in Figure 6 use batch normalization layers, as recommended for ResNet-like architectures. In this case, MIP still provides a substantial improvement in terms of model convergence and training stability.

**Feature Normalization** - Feature normalization techniques such as layer normalization, instance normalization or group normalization do remove the proportionality phenomenon we identify (Ba et al., 2016; Ulyanov et al., 2016). However, by doing so they make the predicted weights independent of the input hyperparameter, limiting the modeling capacity of the hypernetwork architecture. Moreover, in our empirical analysis, networks with layer normalization in the hypernetwork layers failed to train entirely, with the loss diverging early in training.

**Weight Normalization** - We also considered techniques that decouple the gradient magnitude and direction such as weight normalization (Qiao et al., 2019). Performing weight normalization on the hypernetwork predictions effectively decouples the gradient magnitude and direction. We find that convergence is substantially lower compared to the default parametrization. Moreover, final model performance does not match the default parametrization.

