# OpenReview forum: "Magnitude Invariant Parametrizations Improve Hypernetwork Learning"
_ICLR.cc/2024/Conference — ICLR 2024 poster_

### Official Review · Reviewer_YXs9 · 2023-10-30

**Soundness:** 3 good
**Presentation:** 3 good
**Contribution:** 2 fair
**Rating:** 5
**Confidence:** 4

**Summary:**

This paper identifies a problem that contributes to the challenge of training for hypernetworks: The magnitude proportionality between the inputs and outputs of the hypernetwork. The authors demonstrate how this proportionality can result in unstable optimization processes. To mitigate this issue, they introduce a magnitude-invariant parameterization method for hypernetworks, demonstrating its effectiveness in stabilizing the training process and accelerating convergence.

**Strengths:**

- The techniques proposed in this paper are indeed helpful in stabilizing the training of hypernetworks. Given that hypernetworks are widely used in machine learning. These techniques can be useful for practitioners.

**Weaknesses:**

- The discussion regarding why input/output proportionality results in unstable training is somewhat insufficient. It is a well-known fact that neural networks with piecewise linear activation functions, such as ReLU networks, exhibit this proportionality between inputs and outputs. However, if proportionality is the main reason, why ReLU networks are not well-known for having such problems for classical tasks such as regression/classification (not in hypernetworks)? Therefore, I think more explanations or discussions are needed in the text.
- The two strategies introduced in the paper, namely input encoding and additive output formulation, are a bit ad hoc from my perspective. While these methods undoubtedly offer practical benefits for hypernetworks, I am not sure if the contribution is significant enough. A more thorough exploration and explanation of the theoretical underpinnings that underscore the significance of these techniques would enhance the paper's contribution and provide a clearer justification for their adoption.

**Questions:**

- I might miss the related information. Apologies if that is the case. Are there any results of combining additive output with BatchNorm/LayerNorm in Figure 5? It is interesting to see such results because it verifies that MIP has something that normalization layers cannot offer. If proportionality is the main reason, what are the theoretical reasons that MIP can solve the problem but normalization cannot?
- If the problem is proportionality, how about we use other activation functions such as ELU or GELU? Does that solve the problem? If not, can we still say the problem is proportionality?

---

> ### Author Response · Authors · 2023-11-16
>
> We appreciate the constructive feedback and comments. Addressing the
> raised questions:
>
> **Weaknesses:**
>
> > The discussion regarding why input/output proportionality results in
> > unstable training is somewhat insufficient. It is a well-known fact
> > that neural networks with piecewise linear activation functions, such
> > as ReLU networks, exhibit this proportionality between inputs and
> > outputs. However, if proportionality is the main reason, why ReLU
> > networks are not well-known for having such problems for classical
> > tasks such as regression/classification (not in hypernetworks)?
> > Therefore, I think more explanations or discussions are needed in the
> > text.
>
> This is a natural question, that we should have addressed more directly.
> We will revise the manuscript to include further discussion about why
> the input/output proportionality issue is less important for regular
> neural networks. Most notably:
>
> -   Deep neural networks are often used to model high dimensional
>     unstructured data such as image, text, or audio. In these settings,
>     neural network inputs are most often standardized to have zero mean
>     and unit variance, which reduces the effect of the proportionality
>     issue. In contrast, many applications of hypernetworks have
>     relatively low input dimensionality. For example, hypernetworks are
>     often used with scalar or low dimensional inputs. The
>     proportionality issue is exacerbated in hypernetwork problems that
>     feature low dimensional or scalar input spaces.
>
> -   A similar issue has been demonstrated for some applications of
>     regular networks, like NeRFs and implicit networks. In such cases,
>     alternatives such as sinusoidal encodings and activation functions
>     have been shown to help \[1,2\].
>
> \[1\] Implicit Neural Representations with Periodic Activation Functions  - Sitzmann et al.
>
> \[2\] Fourier Features Let Networks Learn High Frequency Functions in
> Low Dimensional Domains - Tancik et al.
>
> **Questions:**
>
> > I might miss the related information. Apologies if that is the case.
> > Are there any results of combining additive output with
> > BatchNorm/LayerNorm in F igure 5? It is interesting to see such
> > results because it verifies that MIP has something that normalization
> > layers cannot offer. If proportionality is the main reason, what are
> > the theoretical reasons that MIP can solve the problem but
> > normalization cannot?
>
> The reviewer is correct, Figure 5 does not include results combining the
> additive output with BatchNorm/LayerNorm. In our experiments, adding
> normalization layers to MIP had little effect in per-epoch convergence,
> but it increased the per-epoch computational cost because of the
> additional overhead of the normalization layers.
>
> More generally, the fundamental reason normalization layers are not well
> suited to deal with this issue is that they remove degrees of freedom by
> enforcing zero mean and unit variance in the input. Since hypernetworks
> often feature low dimensional input spaces, removing degrees of freedom
> in the input introduces substantial information loss. Enforcing constant
> magnitude inputs has similar shortcomings. This is why our input
> encoding maps each dimension to a separate pair of coordinates.
>
> > If the problem is proportionality, how about we use other activation
> > functions such as ELU or GELU? Does that solve the problem? If not,
> > can we still say the problem is proportionality?
>
> We found that other activations like GELU, SiLU/Swish and Tanh, while
> not having the same proportionality property, still propagate the input
> magnitude enough for it to be detrimental in training. MIP improves
> training for all choices of non-linear activation. See section C.3 of
> the supplement.

---

### Official Review · Reviewer_7ccS · 2023-10-30

**Soundness:** 3 good
**Presentation:** 4 excellent
**Contribution:** 3 good
**Rating:** 8
**Confidence:** 3

**Summary:**

This work proposes a novel parametrisation for hypernetworks that is magnitude invariant (MIP). The main motivation for MIP stems from the author’s observation that the hypernetwork output, when using piece-wise linear activations, has a magnitude proportional to the hypernetwork input. The authors argue that this proportionality is detrimental for the hypernetwork optimization as it affects the gradient variance. MIP is then introduced as a way to remove this dependence on the magnitude by parametrising the input to the hypernetwork in terms of Fourier features that have a constant norm throughout training. Then, in order to allow for hypernetwork outputs that are non-proportional to the hypernetwork inputs, the authors propose a residual parametrisation where an auxiliary weight matrix is trained directly and the hypernetwork output is used as an additive correction to that weight matrix. The authors then show that these two modifications to standard hypernetwork training, solve the proportionality problem and improve hypernetwork performance across the board.

**Strengths:**

- The authors identify a novel issue that seems to be important (based on the improvement delta from MIP) for hypernetwork training.
- The specific MIP parametrisation is novel and practically broadly useful for any task that involves hypernetworks.
- The experiments are relatively extensive in terms of tasks and ablation / robustness studies.
- The paper is mostly well written and clear in the presentation of the main ideas and results.

**Weaknesses:**

- While the authors do empirically show that MIP benefits training, it is not clear whether the increased variance could also be controlled with, e.g., appropriately chosen (i.e., lower) learning rates and (i.e., higher) momentum, in the original parametrisation (which could attain similar performance, albeit slower).
- This is something that the authors themselves identify, but given that hypernetworks are becoming popular for fast adaptation of pertained models, e.g., [1], it is important to see whether the magnitude proportionality effect is also detrimental there.

[1] HyperDreamBooth: HyperNetworks for Fast Personalisation of Text-to-Image Models, Ruiz et al., 2023

**Questions:**

I find the overall paper to be well written, the arguments clear and the proposed solution convincing. Therefore, I am happy to recommend acceptance. My questions and suggestions are the following:
- It would be interesting to see whether the residual formulation of the hypernetwork closes meaningfully the gap between fully task specific parameters and the ones predicted by the hyper network, i.e., $\theta^0 + h(E_{L2}(\gamma); w)$. For example, what is the performance if on a new task $t$ one starts from $\theta^0$ and just optimises for a specific number of steps on that task to get $\theta_t^*$? Is $\theta_t^* - \theta_0$ related to $h(E_{L2}(\gamma); w)$? Is the performance of $\theta_t^*$ similar to the performance of $\theta^0 + h(E_{L2}(\gamma); w)$?
- Does the update given by the hypernetwork in this residual formulation need to be dense? Is the hypernetwork only adapting a few dimensions of $\theta^0$?
- It is not clear why for non-scalar inputs $\gamma$, i.e., $\gamma \in \mathbb{R}^D$ with $D \geq 2$  a simple unit normalisation transformation, i.e., $\hat{\gamma} = \frac{1}{\|\|\gamma\|\|_2}\gamma$ would not work for removing the dependence on the magnitude. It seems to me that in this case each $\hat{\gamma}$ would just correspond to a different point on the hypersphere and the output of the hypernetwork would not be independent of the input.

---

> ### Author Response · Authors · 2023-11-16
>
> We are thankful for the detailed, careful, and thought-provoking
> constructive feedback and comments. Addressing the raised questions:
>
> **Weaknesses:**
>
> > While the authors do empirically show that MIP benefits training, it
> > is not clear whether the increased variance could also be controlled
> > with, e.g., appropriately chosen (i.e., lower) learning rates and
> > (i.e., higher) momentum, in the original parametrisation (which could
> > attain similar performance, albeit slower).
>
> This is an important point. In our experiments, we controlled for this
> by testing a range of learning rates and reporting the best converging
> models. We consistently found that MIP was less sensitive to
> hyperparameter choices than the default formulation, and that there were
> settings, like HyperMorph with momentum SGD, where we could not find any
> suitable hyperparameter setting (learning rate & momentum factor) that
> would allow the model to meaningfully train.
>
> **Questions:**
>
> The reviewer raises many interesting questions and suggests interesting
> areas of future work. Regrettably, we do not have the time or space to
> address many of these for this submission in sufficient detail, but we
> are thankful for the suggestions as they will inform future work.
>
> > It would be interesting to see whether the residual formulation of the
> > hypernetwork closes meaningfully the gap between fully task specific
> > parameters and the ones predicted by the hyper network, i.e.,
> > $\theta^0 + h(E_{L2}(\gamma);\omega)$. For example, what is the
> > performance if on a new task one starts from $\theta^0$ and just
> > optimizes for a specific number of steps on that task to get
> > $\theta^\ast_t$? Is $\theta^\ast_t - \theta^0$ related to
> > $h(E_{L2}(\gamma);\omega)$. Is the performance of $\theta^\ast_t$
> > similar to the performance of $h(E_{L2}(\gamma);\omega)$ ?
>
> While not exactly the same as the setup described by the reviewer, we
> did perform experiments where we individually trained classical networks
> for each task $\theta_t$ for the HyperMorph and SSHN settings, to
> understand how classical networks converged for these settings. When
> comparing the task-specific networks to the performance of the
> hypernetwork weights for the same task, we found that hypernetwork
> weights performed no worse than task-specific weights.
>
> > Does the update given by the hypernetwork in this residual formulation
> > need to be dense? Is the hypernetwork only adapting a few dimensions
> > of $\theta^0$?
>
> In our experiments, the contribution of the hypernetwork $\Delta\theta$
> was not particularly sparse, but perhaps this is because the learning
> objective was not set up to promote this property in the first place. We
> believe that adding a sparsity constraint either to the loss, or
> structurally (such as with low rank decompositions) would be an
> interesting avenue for future experiments. We will include this
> discussion in the revised manuscript.
>
> > It is not clear why for non-scalar inputs $\gamma$, i.e.,
> > $\gamma \in \mathbb{R}^D$ with $D \geq 2$ a simple unit normalisation
> > transformation, i.e., $\hat{\gamma} = 1/||\gamma||\gamma$ would not
> > work for removing the dependence on the magnitude. It seems to me that
> > in this case each $\hat{\gamma}$ would just correspond to a different
> > point on the hypersphere and the output of the hypernetwork would not
> > be independent of the input.
>
> This is a good question that we should have addressed more directly.
> Dividing by the norm of the input produces information loss in the
> hypernetwork input space, even for multi-dimensional hypernetwork
> inputs. As an example, we can consider inputs $\gamma_A = [0.1, 0.2]$
> and $\gamma_B = [0.4, 0.8]$, dividing them by their respective norms
> results in
> $\gamma_A / ||\gamma_A|| = \gamma_B / ||\gamma_B|| = [0.447,0.894]$.
> While $\gamma_A$ and $\gamma_B$ correspond to different points in the
> input space, they are mapped to the same constant norm representation.
> This is why our input encoding maps each dimension to a separate pair of
> coordinates.
>
> Since hypernetworks often feature low dimensional input spaces, removing
> a degree of freedom in the input introduces substantial information
> loss.

---

> > ### Comment · Reviewer_7ccS · 2023-11-22
> > **Response to rebuttal**
> >
> > I thank the authors for their response and encourage them to upload the revised manuscript with the updated discussions. Finally, one more point perhaps worthwhile to discuss is the dependence on the dimensionality. While the authors mention that HyperNetworks typically use low-dimensional representations, this is a modelling choice and in general one is free to choose any dimensionality (as opposed to standard networks where the dimensionality of the input data is given and fixed). Therefore, given this available degree of freedom, it is unclear whether Fourier features (which themselves double the input dimensionality) or simple normalization methods (that maintain the same input dimensionality) are better.

---

### Official Review · Reviewer_6gdA · 2023-11-01

**Soundness:** 3 good
**Presentation:** 3 good
**Contribution:** 3 good
**Rating:** 6
**Confidence:** 3

**Summary:**

The paper presents a solution, Magnitude Invariant Parametrizations (MIP), to a previously unidentified optimization problem in hypernetwork training that causes large gradient variance and unstable training dynamics. MIP, by modifying the typical hypernetwork formulation, addresses this issue without adding training or inference costs. The authors extensively test MIP, demonstrating improved stability and faster convergence in hypernetwork training across various settings. They also release HyperLight, an open-source PyTorch library, to ease the implementation of MIP and promote hypernetwork adoption. Through rigorous analysis and experimentation, the paper showcases MIP's potential in substantially enhancing hypernetwork training, marking a significant advancement in this domain.

**Strengths:**

The strengths of this paper include the identification of a novel optimization problem in hypernetwork training, the proposal of a new formulation (MIP) that addresses this issue without extra computational costs, extensive testing and comparative analysis demonstrating MIP's effectiveness, and the provision of an open-source library, HyperLight, to facilitate the practical adoption of the proposed solution in hypernetwork models. Through rigorous analysis and extensive experimentation, the paper makes a significant contribution towards improving the stability and convergence speed in hypernetwork training, providing a promising direction for the community.

**Weaknesses:**

The paper mainly focuses on fully connected layers and common activation, initialization choices, and optimizers (SGD with momentum and Adam) in its experiments, which may not encompass a broader spectrum of hypernetwork architectures or other types of networks. There's also a mention of unexplored territories like the effect of MIP on transfer learning and other less common architectures and optimizers, indicating a scope for broader empirical validation. Furthermore, the impact of MIP on real-world applications or larger-scale problems is not thoroughly explored, which might be crucial for the adoption of this technique in practical settings.

**Questions:**

Mentioned in the weakness.

---

> ### Author Response · Authors · 2023-11-16
>
> We appreciate the constructive feedback and comments. Addressing the
> raised weaknesses:
>
> > The paper mainly focuses on fully connected layers and common
> > activation, initialization choices, and optimizers (SGD with momentum
> > and Adam) in its experiments, which may not encompass a broader
> > spectrum of hypernetwork architectures or other types of networks.
>
> We extensively tested our proposed parametrization across learning
> tasks, primary model architectures, optimizers, initialization, and
> non-linear activations, finding consistent improvements in all settings.
> That said, we agree with the reviewer that hypernetwork architectures
> and applications are diverse and varied. In our work, we tried to cover
> the settings that are most prevalent in the literature. Are there
> specific architectures that the reviewer would like us to try?
>
> > There's also a mention of unexplored territories like the effect of
> > MIP on transfer learning and other less common architectures and
> > optimizers, indicating a scope for broader empirical validation.
>
> We agree that other applications are an interesting avenue for future
> work. Given that we see different (though not directionally different)
> results for Adam and SGD (which together dominate the literature), it
> would be interesting to see what happens with other optimizers that
> might become available.
>
> > Furthermore, the impact of MIP on real-world applications or
> > larger-scale problems is not thoroughly explored, which might be
> > crucial for the adoption of this technique in practical settings.
>
> We agree that evaluation on real world data is important. We did show
> the benefits that MIP brings to practical medical imaging applications.
> The HyperMorph task of loss-regularization for medical image
> registration features a complex medical imaging task and solves a real
> world problem.

---

### Official Review · Reviewer_mEvR · 2023-11-01

**Soundness:** 2 fair
**Presentation:** 2 fair
**Contribution:** 3 good
**Rating:** 6
**Confidence:** 3

**Summary:**

The authors study the problem of improving the stability and efficiency of training hypernetworks. They first observe a previously unidentified problem with hypernetwork training, namely that for certain hypernetwork architectures the scaling of the input to the hypernetwork leads to the proportional scaling of the outputs. This can lead to destabilization of the training procedure and slow training. The authors propose a a novel hypernetwork formulation that removes this proportionality property. Across several architectures, tasks and optimizers, their framework leads to improved training stability and convergence.

**Strengths:**

While normalizing inputs to neural network models is already well established best practice, to the best of my knowledge the specific application of this best practice for hypernetwork inputs has not been studied as much. I consider the fact that the input encoding approach of the authors is straightforward a plus. The experimental validation of the key claims of the papers is extensive.

**Weaknesses:**

I am not really sure if the output encoding part of the framework fits well with the problem that the authors claim to solve. It is not clear how output encoding relates to the input and output proportionality problem. It also makes interpreting the experimental results where both input and output encoding are used harder. Intuitively, output encoding allows the model to learn the task even if the hypernetwork does nothing so it is not clear if we improve hypernetwork training or just make the hypernetwork path less critical and rely on "classical" parameter learning which is typically more stable.

Another thing that is not clear to me is that it is not clear to me what is special about hypernetworks so that input normalization needs to work differently than "classical" networks. The scaling property would hold for a "classical" network as well for the appropriate activations. And yet Appendix C.5 claims that standard normalization techniques of "classical" networks would not work for hypernetworks. Appendix C5 seems to suggest that "classical" normalization techniques could make the output independent of the input. It is not clear to me why this is a problem only for hypernetworks and not for "classical" networks.

Also it would be useful to clarify why one could not just divide the hypernetwork input by its norm, at least for the case of a multi-dimensional hypernetwork inputs. In my mind, if the problem was just the scaling of the output, this should have worked. The suggested methods go beyond just fixing the scale of the input vector as a whole so it is not very clear if the problem is indeed the input output proportionality or some more general feature scaling issue.

**Questions:**

I have summarized above some points that were not very clear to me. I would be willing to increase my score if they were to be clarified.

---

> ### Author Response · Authors · 2023-11-16
>
> We appreciate the constructive feedback and comments. Addressing the
> raised questions:
>
> > I am not really sure if the output encoding part of the framework fits
> > well with the problem that the authors claim to solve. It is not clear
> > how output encoding relates to the input and output proportionality
> > problem. It also makes interpreting the experimental results where
> > both input and output encoding are used harder. Intuitively, output
> > encoding allows the model to learn the task even if the hypernetwork
> > does nothing, so it is not clear if we improve hypernetwork training
> > or just make the hypernetwork path less critical and rely on
> > \"classical\" parameter learning which is typically more stable.
>
> This is subtle, and we should have done a better job of explaining it,
> and will do so. Using the output encoding changes the learning problem
> for the hypernetwork. Rather than learning how to predict parameters,
> the output encoding provides a task-independent parameter initialization
> so that the hypernetwork only has to learn the delta for each task.
> Under the default formulation, the prior is that there is no commonality
> between tasks. With the output encoding, we change that prior so that
> there is some commonality.
>
> In Figure 5 we include an ablation of input and output encoding, where
> we find that each component improves the learning process individually,
> and that best results are achieved when both are used jointly.
>
>
> > Another thing that is not clear to me is that it is not clear to me
> > what is special about hypernetworks so that input normalization needs
> > to work differently than \"classical\" networks. The scaling property
> > would hold for a \"classical\" network as well for the appropriate
> > activations. And yet Appendix C.5 claims that standard normalization
> > techniques of \"classical\" networks would not work for hypernetworks.
> > Appendix C5 seems to suggest that \"classical\" normalization
> > techniques could make the output independent of the input. It is not
> > clear to me why this is a problem only for hypernetworks and not for
> > \"classical\" networks.
>
> Our observation that "normalization techniques could make the output
> independent of the input" was referring to the specific case of scalar
> inputs. We should have said this, and we will revise the manuscript to
> reflect that.
>
> Including normalization layers does make activations follow normalized
> distributions, but they introduce learning challenges in settings with
> low dimensional inputs. The reason why these challenges rarely arise
> with classical networks is that classical networks are rarely used with
> low dimensional or scalar input spaces. In contrast, hypernetworks are
> often used to predict parameters based on hyperparameters of interest,
> and frequently have low dimensional or scalar inputs.
>
> > Also it would be useful to clarify why one could not just divide the
> > hypernetwork input by its norm, at least for the case of a
> > multi-dimensional hypernetwork inputs. In my mind, if the problem was
> > just the scaling of the output, this should have worked. The suggested
> > methods go beyond just fixing the scale of the input vector as a whole
> > so it is not very clear if the problem is indeed the input output
> > proportionality or some more general feature scaling issue.
>
> This is a good question that we should have addressed more directly.
> Dividing by the norm of the input produces information loss in the
> hypernetwork input space, even for multi-dimensional hypernetwork
> inputs. As an example, we can consider inputs $\gamma_A = [0.1, 0.2]$
> and $\gamma_B = [0.4, 0.8]$, dividing them by their respective norms
> results in
> $\gamma_A / ||\gamma_A|| = \gamma_B / ||\gamma_B|| = [0.447,0.894]$.
> While $\gamma_A$ and $\gamma_B$ correspond to different points in the
> input space, they are mapped to the same constant norm representation.
> This is why our input encoding maps each dimension to a separate pair of
> coordinates.

---

> > ### Comment · Reviewer_mEvR · 2023-11-22
> > **I appreciate the response**
> >
> > I appreciate the response. and I think the suggested improvements to the manuscript will indeed help.
> >
> > Based on the author's comments, I am starting to see what is the hyper-network specific problem, mainly that input normalization is  more important when the input space is low dimensional. In contrast, magnitude proportionality seems a bit less convincing since the improvements by input normalization are there even for bounded activation functions. I am not sure if using magnitude proportionality as motivation is actually helping.
> >
> > Regarding output encoding, I understand the motivation of the encoding and I thank the authors for the explanations. I agree that output encoding changes the prior of the learning problem. I still remain uncertain about the existence of a connection between magnitude proportionality and output encoding.  The input and output encoding modifications could have been two different papers with minimal overlap beyond hypernetwork basics.
> >
> > All in all, I feel the pitch of this work needs improvement. But I lean towards acceptance so I increase my rating to a 6.

---

### Meta-Review · Area_Chair_rJic · 2023-11-27

**Metareview:**

The reviewers and meta reviewer all carefully checked and discussed the rebuttal. They thank the authors for their response and their efforts during the rebuttal phase.

The reviewers and meta reviewer all acknowledge the sound contribution of identifying and carefully studying a current weakness of hypernetworks. Similarly, the reviewers and meta reviewer all call out the overall good quality of the manuscript (writing, structure).
However, there are still some remaining important concerns. As a result, the reviewers and the meta reviewer are weakly inclined to accept the paper.

In particular, the authors are urged to carefully update their final manuscript with the following points:

* Tighten up the motivation and the narrative around the magnitude proportionality, the output encoding and their possible relationship (mEvR, YXs9). On a related note, discuss the related work [A] whose output encoding is reminiscent of that proposed here.
* Add the good studies suggested by 7ccS, namely (i) Does the residual formulation of the hypernetwork close meaningfully capture fully task-specific parameters and the ones predicted by the hypernetwork? (see proposed experiment in the review)  and (ii) Fourier features vs. simple normalization methods? (“...removing a degree of freedom in the input introduces substantial information loss…” really manifests in practice?)

If the paper was submitted to a journal, it would be accepted conditioned on those key changes, the meta reviewer thus expects all those changes to be carefully implemented.

[A] Bae & Grosse, Neurips 2020 [paper link](https://proceedings.neurips.cc/paper_files/paper/2020/file/f754186469a933256d7d64095e963594-Paper.pdf)

**Justification For Why Not Higher Score:**

* The narrative about the two ingredients proposed to solve the identified problem needs some improvement
* The problem tackled by the paper may not impact the broader ICLR community
* Some theoretical support would have been beneficial

**Justification For Why Not Lower Score:**

* Clear and well-presented manuscript
* Clearly identified problem with hypernetworks
* Rigorous and convincing experiments

---

### Decision · Program_Chairs · 2024-01-16

Accept (poster)